# Differentially Private Multi-Armed Bandits in the Shuffle Model

Jay Tenenbaum[*]        Haim Kaplan[†]        Yishay Mansour[‡]        Uri Stemmer[§]

## Abstract

We give an $(\varepsilon, \delta)$-differentially private algorithm for the multi-armed bandit (MAB) problem in the shuffle model with a distribution-dependent regret of $O\left(\left(\sum_{a \in [k]: \Delta_a > 0} \frac{\log T}{\Delta_a}\right) + \frac{k\sqrt{\log \frac{1}{\delta}} \log T}{\varepsilon}\right)$, and a distribution-independent regret of $O\left(\sqrt{kT \log T} + \frac{k\sqrt{\log \frac{1}{\delta}} \log T}{\varepsilon}\right)$, where $T$ is the number of rounds, $\Delta_a$ is the suboptimality gap of the arm $a$, and $k$ is the total number of arms. Our upper bound almost matches the regret of the best known algorithms for the centralized model, and significantly outperforms the best known algorithm in the local model.

## 1   Introduction

The *multi-armed bandit (MAB)* problem is a classical sequential decision-making problem in which an agent tries to maximize a cumulative stochastic reward [27, 23] under uncertainty. This problem, which is applicable to various areas such as recommender systems, online advertising and clinical trials, embodies the well known exploration-exploitation trade-off between learning the environment and acting optimally based on our current knowledge about the environment.

More formally, in the MAB problem at each time $t = 1, \ldots, T$ an agent chooses an arm $i$ from the set $[k] = \{1, \ldots, k\}$ of $k$ arms, and obtains an iid reward $r^t$ drawn from the unknown distribution $R_i$ over $\{0, 1\}$ with expectation $\mu_i = \mathbb{E}[R_i]$. Let $a^* = \arg\max_a \mu_a$ be an arm with the largest expected reward, and denote this reward by $\mu^* = \mu_{a^*}$. Let the (suboptimality) gap of an arm $a$ to be the gap between its expected reward and that of $a^*$, i.e., $\Delta_a = \mu^* - \mu_a$. The agent's goal is to maximize the total expected reward, or rather to minimize the expected regret $R(T) = T \cdot \mu^* - \mathbb{E}\left[\sum_{i=1}^{T} r^t\right]$ defined to be the expected gap between the algorithm and the optimal algorithm that knows the distributions $R_i$.

In this work we address the privacy in such a setting. As a motivating example, consider an advertisement system in which the server presents to each user an advertisement $a \in [k]$. The user then decides whether to click on the advertisement or not. This click decision depends on different private characteristic of the user. The user then reports to the server whether it clicked on the advertisement (in which case its reward is $r = 1$) or not ($r = 0$). From this example, it is clear that $r$ is private information of the user, and using traditional algorithms for the MAB problem incautiously might leak user-private data.

In order to mathematically alleviate privacy concerns, Dwork et al. [11] defined the notion of *differential privacy (DP)*, which requires that the output of the computation has a limited dependency

---

[*]Google Research. jayten@google.com.

[†]Blavatnik School of Computer Science, Tel Aviv University and Google Research. haimk@tau.ac.il.

[‡]Blavatnik School of Computer Science, Tel Aviv University and Google Research. mansour.yishay@gmail.com.

[§]Blavatnik School of Computer Science, Tel Aviv University and Google Research. u@uri.co.il.

35th Conference on Neural Information Processing Systems (NeurIPS 2021).

Table 1: Best-known MAB regret upper and lower bounds for various DP models.

| Privacy model | Best-known regret upper and lower bounds[6] |
|---|---|
| **Centralized** $(\varepsilon, 0)$**-DP** | $\Theta\left(\left(\sum_{a \in [k]: \Delta_a > 0} \frac{\log T}{\Delta_a}\right) + \frac{k \log T}{\varepsilon}\right)$ [24, 25] |
| **Centralized** $(\varepsilon, \delta)$**-DP** | $O\left(\left(\sum_{a \in [k]: \Delta_a > 0} \frac{\log T}{\Delta_a}\right) + \frac{k}{\varepsilon}\right)$ [28] |
| **Local** $(\varepsilon, 0)$**-DP** | $\Theta\left(\frac{1}{\varepsilon^2} \sum_{a \in [k]: \Delta_a > 0} \frac{\log T}{\Delta_a}\right)$ [22][7] |
| **Shuffle** $(\varepsilon, \delta)$**-DP (ours)** | $O\left(\left(\sum_{a \in [k]: \Delta_a > 0} \frac{\log T}{\Delta_a}\right) + \frac{k\sqrt{\log \frac{1}{\delta}} \log T}{\varepsilon}\right)$ |

on any single user's data. Formally, a mechanism $(\varepsilon, \delta)$-DP if for any pair of neighboring inputs (differing by a single user's data), the probability that the mechanism outputs a value in any set $B$ is not different by more than a multiplicative factor of $e^\varepsilon$ and an additive factor of $\delta$. *Differential privacy* has been extensively studied under many different sub-models of privacy. On one end of the spectrum lies the *centralized model* of differential privacy, where the users trust the server with their data, and the liability to protect user privacy lies on the server, who must make sure that any data published externally (e.g., aggregated statistics) respects the privacy constraints. On the other end of the spectrum lies the (strictly stronger) *local model* of differentialy privacy (LDP), where the user privatizes its own data prior to sending it to the server.[5]

Differentially private versions of the MAB problem have been considered in various previous works, where the private information are users' rewards (two neighboring inputs differ by the reward value of a single user), and the algorithm's output is the subsequent arm(s) it selects. Table 1 summarizes the best known distribution-dependent regret bounds for the various privacy models, together with our new result in the *shuffle model* which we soon define formally.

With real-world algorithms gradually moving away from the *centralized model* of privacy, the immediate question is "can we design a private algorithm for MAB with privacy guarantees which are similar to local DP, but with similar regret to centralized DP (without a multiplicative $1/\varepsilon^2$ factor)?".

To address the inevitable gap between the local and centralized models, which is in fact common in the literature of differential privacy, the alternative *shuffle model* [6, 9, 13] explores the space in between the local and centralized models by introducing a trusted shuffler that receives user messages and permutes them (i.e., disassociates a message from its sender) before they are delivered to the server. For privacy analysis, we assume that the shuffle is perfectly secure, i.e., its output contains no information about which user generated each of the messages. This is traditionally achieved by the shuffler stripping implicit metadata from the messages (e.g., timestamps, routing information), and frequently forwarding this data to remove time and order information. The shuffle model ensures that sufficiently many reports are collected in each round so that any one report can hide in a shuffled batch. In order to apply the shuffle model to the MAB problem in the context of advertisements, we divide the algorithm into batches, where before each batch we decide on the fly its size $m$, and then present the $m$ next users the same advertisement $a$, and finally apply a private shuffle model mechanism to their rewards to communicate reward aggregate information to the server.[8]

A constantly growing body of work presents new and improved mechanisms in the shuffle model for basic statistical tasks [16, 13, 12, 2], such as *private binary summation*, in which the server must privately approximate the sum of a collection of values $x_1, ..., x_m \in \{0, 1\}$ held by the $m$ users in

---

[5]Formally, in the non-interactive setting, a mechanism is $(\varepsilon, \delta)$-LDP if for any two user inputs, the probability that the privatizer sends the server a value in any set $B$ is not different by more than a multiplicative factor of $e^\varepsilon$ and an additive factor of $\delta$.

[6]The corresponding distribution-independent regret bounds usually simply replace the $\sum_{a \in [k]: \Delta_a > 0} \frac{\log T}{\Delta_a}$ term with $\sqrt{kT \log T}$.

[7]This lower bound can be extended from $(\varepsilon, 0)$-LDP to $(\varepsilon, \delta)$-LDP using arguments from Bun et al. [7] and Cheu et al. [9] since we focus on single-round (non-interactive) mechanisms in which the user can only send information to another party once.

[8]We remark that a given user does not know in advance the size of the batch in which it participates, since this size depends on the algorithm's run.

the shuffle. For *private binary summation*, the optimal achievable errors in the central, local and shuffle model are $\tilde{\Theta}(1/\varepsilon)$ [11], $\tilde{\Theta}(\sqrt{m}/\varepsilon)$ [5, 8] and $\tilde{\Theta}(1/\varepsilon)$ [9] respectively, where the $\tilde{\Theta}(\cdot)$ hides poly-logarithmic terms. Similar errors hold for the *private summation* problem which approximates the sum of real values in $[0, 1]$.

## 1.1 Our contributions

To the best of our knowledge, our work is the first to consider the MAB problem under the shuffle model of differential privacy. In order to support the online nature of the MAB problem, we consider a variant of the shuffle model. As opposed to the classical shuffle model, in which the shuffle size is unbounded and the mechanism runs only once, we continuously run shuffle mechanisms for many disjoint batches of users who can only afford a single round of communication.[9]

We consider the paradigm where the server controls the different users who are cooperative and communicate only with the server. We give a rigorous definition of *shuffle differential privacy (SDP)* for the multi-armed bandit problem (assuming binary rewards), and give and prove the first two such algorithms. Our algorithms *Shuffle Differentially Private Arm Elimination (SDP-AE)* and *Variable Batch Shuffle Differentially Private Arm Elimination (VB-SDP-AE)* are both based on the well-known *arm elimination (AE)* algorithm, using consecutive batches of users and together with an SDP private binary summation mechanism.

We show that the simpler but weaker SDP-AE achieves a distribution-dependent regret of $O\left(\left(\sum_{a\in[k]:\Delta_a>0}\frac{\log T}{\Delta_a}\right)+\frac{k\log\frac{1}{\delta}}{\varepsilon^2}\right)$, and a distribution-independent regret of $O\left(\sqrt{kT\log T}+\frac{k\log\frac{1}{\delta}}{\varepsilon^2}\right)$.[10]

We then describe VB-SDP-AE, a generalization of SDP-AE to exponentially growing batch sizes, and prove it has a distribution-dependent regret of $O\left(\left(\sum_{a\in[k]:\Delta_a>0}\frac{\log T}{\Delta_a}\right)+\frac{k\sqrt{\log\frac{1}{\delta}}\log T}{\varepsilon}\right)$, and a distribution-independent regret of $O\left(\sqrt{kT\log T}+\frac{k\sqrt{\log\frac{1}{\delta}}\log T}{\varepsilon}\right)$.

Note that, compared to the local model (Ren et al [22]), the regret of both SDP-AE and VB-SDP-AE is improved, by having the dependency on $1/\varepsilon$ be additive rather than multiplicative. In addition, VB-SDP-AE almost matches the regret of the best known algorithms for the centralized model, that is the distribution-dependent regret of $O\left(\left(\sum_{a\in[k]:\Delta_a>0}\frac{\log T}{\Delta_a}\right)+\frac{k}{\varepsilon}\right)$ of Tossou and Dimitrakakis [28].

## 1.2 Related work

The differentially private MAB problem has been considered in many previous works [19, 17, 21]. Shi and Shen [26] and Dubey and Pentland [10] studied MAB and linear bandits respectively in the federated setting. Zheng et al. [29] studied contextual bandits with LDP. Batched MAB with a predetermined number of batches was studied in [14, 15].

For the private summation problem, Cheu et al. [9] gave unbiased $(\varepsilon, \delta)$-SDP mechanisms over binary inputs and real inputs in $[0, 1]$, with error roughly $\sqrt{\log(1/\delta)}/\varepsilon$ and $\log(1/\delta)/\varepsilon$, respectively. In several works of Balle et al. [2, 3, 1], they gave a biased $(\varepsilon, \delta)$-SDP mechanism for real inputs in $[0, 1]$ with similar error and a constant number of messages, and a single-message $(\varepsilon, \delta)$-SDP mechanism with optimal error.

---

[9]Beimel et al. [4] showed that every centralized-DP mechanism can be emulated in the shuffle model in two (communication-intensive) rounds of communication, however these results are not applicable to our setting since we assume that each online user participates only in one shuffle and then disappears.

[10]In this paper, we assume that $T$ is known apriori. Otherwise, we can apply standard doubling arguments to get roughly the same results.

## 2  Background and preliminaries

### 2.1  Shuffle-model privacy

In the well-studied setting of shuffle model privacy, there are $m$ users, each with data $x_i \in X$. Each user applies some encoder $E : X \to Y^*$ to their data and sends the messages $(y_{i,1}, ..., y_{i,p}) = E(x_i)$ to a shuffler $S : Y^* \to Y^*$. The shuffler then shuffles all the messages $y_{i,j}$ from all the users, and outputs them in a uniformly random order to an analyzer $A : Y^* \to Z$ to estimate some function $f(x_1, ..., x_m)$. Thus, the mechanism $M$ consists of the tuple $(E, S, A)$. We say that such a mechanism $M$ is $(\varepsilon, \delta)$-*shuffle differentially private* (or $(\varepsilon, \delta)$-*SDP* for short) if the output of $S$ is $(\varepsilon, \delta)$-differentially private, or more formally: A mechanism $M = (E, S, A)$ is $(\varepsilon, \delta)$-SDP if for any pair of inputs $\{x_i\}_{i=1}^m$ and $\{x_i'\}_{i=1}^m$ which differ in at most one value, we have for all $B \subseteq Y^*$:

$$P(S(\cup_{i=1}^m E(x_i)) \in B) \leq e^\varepsilon \cdot P(S(\cup_{i=1}^m E(x_i')) \in B) + \delta.$$

In the mechanism used in this paper, $E$ outputs the user's reward bit together with a set of random bits, and $A$ sums all these bits and debiases the result to get an unbiased estimate of the sum of the users' rewards.

### 2.2  Shuffle-model MAB

Algorithms which are private in the shuffle model typically apply the mechanism $M$ once over a set of $m$ users. Here we study the MAB problem which is an online problem, often deployed in real-world applications and with users which are end-devices such as cellphones with a possibly limited or unreliable internet connection. Hence, to adapt the MAB problem to the shuffle model, we batch sequences of consecutive users, and assume that each user can afford a single round of communication, and is never selected more than once.

**Model and objective**   The *shuffle-model MAB* setting involves repeating the following process until the $T$'th player pulls its arm:

1. The server selects a batch size $m$, a *batch* of $m$ random fresh new users, and an $m$-user single-round SDP mechanism $M$.[11] It then picks an arm $a \in [k]$ that all $m$ users of the batch pull. (For concreteness think that the server picks an ad $a \in [k]$ and sends it to the a random batch of $m$ users.)

2. Each user $i$ determines binary reward from pulling the arm $a$.[12] (For concreteness think that each user decides whether to click on the ad or not. This defines the reward related to user $i$, which is $r_i = 1$ if it clicks the ad and $r_i = 0$ otherwise.)

   Since our $m$ users are random, these rewards are a sample of $m$ independent rewards from the distribution $R_a$ associated with arm $a$.

3. The server computes $M(\{r_i\}_{i \in batch})$ using the rewards $r_i$.

The objective is to minimize the (pseudo) regret, which is the expected difference between the sum of the rewards accumulated (over all the users) by the algorithm and the sum of the rewards of the optimal algorithm that apriori knows an arm with the largest expected reward $a^*$. Let $\mu_a = \mathbb{E}[R_a]$ be the *expected reward* (or simply *mean*) of the arm $a$, let $\mu^*$ denote the expected reward of $a^*$, let $\Delta_a = \mu^* - \mu_a$ be the (suboptimality) gap of the arm $a$ which quantifies the gap between its expected reward and that of $a^*$, and let $N_a$ be the random variable which counts the total number of times the arm $a$ was pulled during the run of the algorithm.

Formally, the (pseudo) regret of the algorithm for $T \in \mathbb{N}$ users is defined to be

$$R(T) = T \cdot \mu^* - \mathbb{E}\left[\sum_{i=1}^T r_i\right] = \mathbb{E}\left[\sum_{a \in [k]} N_a \Delta_a\right].$$

---

[11]Note that always selecting $m = 1$ reduces this setting to the Local model MAB.

[12]For simplicity, we assume that the rewards are binary. However, our algorithms and proofs naturally extend to the real $[0, 1]$ reward setting, by replacing our private binary summation mechanism (defined later, see Appendix C) with a private summation mechanism for real numbers in $[0, 1]$ with similar guarantees.

**Privacy**   Since the private data of each user is its reward, the appropriate adaptation of shuffle model privacy for the multi-armed bandit problem is as follows. An algorithm for the multi-armed bandits problem is $(\varepsilon, \delta)$-*shuffle differentially private (SDP)* if for any batch of users, the shuffle mechanism that we apply over them in step 3 is $(\varepsilon, \delta)$-SDP with respect to the rewards of the users (as we recall, the reward of a user – whether it clicked on an ad or not – depends on its private features). Formally, for every batch we run a shuffle mechanism where the $m$ users are the users of the current batch, and the data $x_i$ of each user is its reward $r_i$, and we require that each such mechanism is $(\varepsilon, \delta)$-SDP.

## 2.3   Concentration bounds

We use the following standard definitions of Sub-Gaussian random variables and Hoeffding's inequality.

**Definition 1** (Sub-Gaussian random variable). *A random variable $X$ with mean $\mu$ is called sub-Gaussian with variance $\sigma^2$, i.e., $X \sim SG(\sigma^2)$ if:*

$$\forall \lambda \in \mathbb{R}, \ \mathbb{E}\left[\exp(\lambda(X - \mu))\right] \leq e^{\lambda^2 \sigma^2 / 2}.$$

An equivalent definition shows that if $\forall t > 0, \max\left(P(X - \mu \geq t), P(X - \mu \leq -t)\right) \leq \exp\left(\frac{-t^2}{2\sigma^2}\right)$, then $X$ is sub-Gaussian with variance $\sigma^2$ (up to constant factor). It is well known that if $X_i \sim SG(\sigma_i^2)$ are independent random variables for $i = 1, \ldots, n$, then $\sum_{i=1}^n X_n \sim SG(\sum_{i=1}^n \sigma_i^2)$ and for any $a, b > 0$, $a \cdot X_1 + b \sim SG(a^2 \cdot \sigma_1^2)$. A bounded random variable $X \in [a, b]$ is $SG((b-a)^2/4)$.

Sub-Gaussian random variables satisfy the following concentration bound,

**Lemma 2.1** (Hoeffding's inequality [18]). *Let $\{X_i\}_{i=1}^n$ be independent $SG(\sigma^2)$ random variables, then $P\left(\left|\sum_{i=1}^n X_i - \mathbb{E}\left[\sum_{i=1}^n X_i\right]\right| \geq t\right) \leq 2\exp\left(\frac{-t^2}{2n\sigma^2}\right).$*

# 3   Differentially private MAB in the shuffle model

In this paper, we use the shuffle model to give a private solution to the multi-armed bandit problem, attaining similar privacy guarantees to that of the local privacy model (LDP), without sacrificing the regret. That is, our algorithm almost matches the best known regret in the centralized model of differential privacy. Our algorithms rely on the fact that algorithms for the multi-armed bandit problem take decisions based on sums of rewards received from the users. Hence, we rely on a particularly efficient and accurate mechanism for private binary summation in the shuffle model as a building block in our algorithms. Specifically, for any $\varepsilon, \delta \in (0, 1)$, let $M_{sum}$ be a private binary summation mechanism, which for any number of users (batch size) $m \in \mathbb{N}$, is $(\varepsilon, \delta)$-SDP, unbiased, and has an error distribution which is independent of the input, and is sub-Gaussian with variance $\sigma_{\varepsilon, \delta}^2 = O\left(\frac{\log \frac{1}{\delta}}{\varepsilon^2}\right)$.[13] Note that our notation of $M_{sum}$ does not include $\varepsilon$ and $\delta$ which will always be clear from context or inherited from the algorithm which runs $M_{sum}$. The challenge is to combine $M_{sum}$ with the well studied *arm elimination (AE)* MAB algorithm to get an SDP algorithm for the MAB problem with almost optimal regret.[14]

## 3.1   SDP-AE: Shuffle Differentially Private Arm Elimination

We base our algorithm on the (non-private) *arm elimination (AE)* algorithm for the MAB problem, which informally maintains a set of viable arms (initially set to be $[k]$), and each phase pulls the set of viable arms sequentially. Once a phase ends, we search for arms which are noticeably suboptimal in comparison to some other arm, and we eliminate them from the set of viable arms.

To adapt AE to the shuffle model, in each phase $t$ each arm is pulled not once, but rather by a whole batch of users. Once all the users in the batch pull the arm and receive their reward, we apply

---

[13]Note that our methods and algorithms should work similarly if they were built over other common algorithms for the MAB problem in which batching makes sense, such as the UCB algorithm.

[14]For completeness, in Appendix C we give a complete description and a proof of such a mechanism $M_{sum}$.

the private binary summation mechanism $M_{sum}$ to the batch's rewards, which gives the server an unbiased but noisy estimate of the sum of rewards in the batch. This estimate has two sources of error, which we account for when we compute the upper confidence bound – the empirical error due to sampling the reward function of the arm, and the error due to the private binary summation mechanism $M_{sum}$.

### 3.1.1 Algorithm outline

In the algorithm below which we call *Shuffle Differentially Private Arm Elimination (SDP-AE)*, we update the estimate of the mean reward of each arm $a$ after every batch of users who sample $a$. The algorithm works in phases, and maintains a set of viable arms initially set to be $[k]$. In each phase, for every viable arm we have a single batch of users sampling it. At the end of phase $t$, we denote by $\hat{S}_a^t$ the noisy estimate of the cumulative sum of the rewards from all previous samples of $a$ in the phases $1, \ldots, t$, and denote by $N_a^t$ the total number of previous samples of arm $a$ in the phases $1, \ldots, t$. The natural estimate for the mean reward of the arm $a$ (denoted by $\hat{\mu}_a^t$) is therefore $\hat{\mu}_a^t = \hat{S}_a^t / N_a^t$. We then calculate the upper and lower confidence bounds $UCB_a^t$ and $LCB_a^t$ respectively of each viable arm $a$ after each phase $t$ using a specific bound which takes into account both sources of error. We finally eliminate any remaining arm with an upper confidence bound which is strictly smaller than the lower confidence bound of some other arm. Algorithm 1 consolidates the algorithm presentation above.

---

**Algorithm 1:** SDP-AE (Shuffle Differentially Private Arm Elimination)

---
1 **Input:** privacy parameters $\varepsilon$ and $\delta$, batch size $m$ and horizon $T$.
2 **Initialize:** $\hat{S}_a^0 = 0$, $N_a^0 = 0$ and $\hat{\mu}_a^0 = 0$ for every $a \in [k]$;
3 Let $\sigma_{\varepsilon,\delta}^2 = O\left(\frac{\log \frac{1}{\delta}}{\varepsilon^2}\right)$ be the sub-Gaussian variance of the error distribution of $M_{sum}$;
4 Let $V = [k]$ denote the set of viable arms;
5 **for** *phase* $t \leftarrow 1, 2, \ldots$ **do**
6     **for** *arm* $a \in V$ **do**
7         **for** *each new user* $i \leftarrow 1$ **to** $m$ **do**
8             User $i$ pulls the arm $a$ and observes reward $r_{a,i}^t$;
9             If total arm samples in current algorithm run is $T$, exit;
10         **end**
11         **Communication:** Perform private binary summation $Z_a^t \leftarrow M_{sum}\left(\{r_{a,i}^t\}_{i=1}^m\right)$;
12         **Server update:** Update $\hat{S}_a^t \leftarrow \hat{S}_a^{t-1} + Z_a^t$, $N_a^t \leftarrow N_a^{t-1} + m$, and finally $\hat{\mu}_a^t \leftarrow \hat{S}_a^t / N_a^t$;
13     **end**
14     **Confidence bounds:** For each arm $a$, calculate $I_a^t \leftarrow \left(\frac{2\sqrt{t}\sigma_{\varepsilon,\delta}}{N_a^t} + \frac{1}{\sqrt{N_a^t}}\right) \cdot \sqrt{2\log T}$, and
        the upper and lower confidence bounds $UCB_a^t \leftarrow \hat{\mu}_a^t + I_a^t$ and $LCB_a^t \leftarrow \hat{\mu}_a^t - I_a^t$;
15     **Elimination:** remove all arms $a$ from $V$ such that $UCB_a^t < \max_{a' \in S} LCB_{a'}^t$;
16 **end**

---

### 3.1.2 Analysis

The privacy is trivial, since each batch we use the $(\varepsilon, \delta)$-SDP mechanism $M_{sum}$. We now focus on regret.

Theorem 3.1 gives a bound on the regret of SDP-AE as a function of the batch size $m$. We follow a somewhat standard regret bound analysis for arm elimination, comprising two parts. The first part uses Hoeffding's inequality to derive a high probability bound on $\left|\hat{\mu}_a^t - \mu_a\right|$, the error between the empirical average reward and the true mean reward of the arm $a$ at a given phase $t$. The second part uses this bound to bound the expected number of times we sample each suboptimal arm $a$, and summing over all suboptimal arms we get a bound on the regret.

**Theorem 3.1.** *The algorithm SDP-AE is $(\varepsilon, \delta)$-SDP, and has a distribution-dependent regret of $O\left(\sum_{a \in [k]:\Delta_a > 0}\left(\frac{\log T}{\Delta_a} + \frac{\sigma_{\varepsilon,\delta}^2 \log T}{m \Delta_a} + m\Delta_a\right)\right)$, and a distribution-independent regret of $O\left(\sqrt{\left(1 + \frac{\sigma_{\varepsilon,\delta}^2}{m}\right) kT \log T} + mk\right)$.*

*Proof sketch.* Fix a phase $t$. For any arm $a$, we have $\hat{\mu}_a^t = \frac{\sum_{s=1}^{t} M_{sum}\left(\{r_{a,i}^s\}_{i=1}^m\right)}{N_a^t}$, where $N_a^t = m \cdot t$. We apply Hoeffding's inequality (Lemma 2.1) once on the sequence of actual rewards of the users, and once on the sequence of errors that $M_{sum}$ introduces into the approximated sum of rewards. We conclude that with high probability, $\forall a \in [k], \forall t \in [T]$, we have $\left|\hat{\mu}_a^t - \mu_a\right| \leq \left(\frac{2\sqrt{t}\sigma_{\varepsilon,\delta}}{N_a^t} + \frac{1}{\sqrt{N_a^t}}\right) \cdot \sqrt{2 \log T}$.

Assuming this high probability event, each arm $a$ is necessarily eliminated after the phase $t_0$ where $\left|\hat{\mu}_a^{t_0} - \mu_a\right|$ becomes at most roughly $\Delta_a/4$. We show that this occurs after at most $O\left(\frac{\log T}{\Delta_a^2} + \frac{\sigma_{\varepsilon,\delta}^2 \cdot \log T}{m\Delta_a^2}\right)$ pulls of the arm $a$, and since we must complete the batch of size $m$, the total regret of the arm $a$ is at most $R_a = O\left(\frac{\log T}{\Delta_a} + \frac{\sigma_{\varepsilon,\delta}^2 \cdot \log T}{m\Delta_a} + m\Delta_a\right)$. To get the distribution-dependent regret we sum the regret above over all arms. For the distribution-independent regret, we split this sum based on if the arm $a$ has $\Delta_a < \beta$ where $\beta = \Theta\left(\sqrt{\frac{(1 + \sigma_{\varepsilon,\delta}^2/m)k \log T}{T}}\right)$ or not, and bound each sum separately. Namely, bound the contribution of arms with $\Delta_a < \beta$ by $\beta T$. □

Recall that the private binary summation mechanism $M_{sum}$'s error distribution is sub-Gaussian with variance $\sigma_{\varepsilon,\delta}^2 = O\left(\frac{\log \frac{1}{\delta}}{\varepsilon^2}\right)$. We fix a concrete batch size $m$ in SDP-AE, and apply Theorem 3.1 to get the following corollary.

**Corollary 3.2.** *SDP-AE with a batch size of $m = \lceil \sigma_{\varepsilon,\delta} \rceil = \Theta\left(\frac{\log \frac{1}{\delta}}{\varepsilon^2}\right)$ is $(\varepsilon, \delta)$-SDP and has a distribution-dependent regret of*

$$O\left(\sum_{a \in [k]:\Delta_a > 0}\left(\frac{\log T}{\Delta_a} + \frac{\Delta_a \log \frac{1}{\delta}}{\varepsilon^2}\right)\right) = O\left(\left(\sum_{a \in [k]:\Delta_a > 0}\frac{\log T}{\Delta_a}\right) + \frac{k \log \frac{1}{\delta}}{\varepsilon^2}\right),$$

*and a distribution-independent regret of $O\left(\sqrt{kT \log T} + \frac{k \log \frac{1}{\delta}}{\varepsilon^2}\right)$.*

### 3.2 VB-SDP-AE: Variable Batch Shuffle Differentially Private Arm Elimination

In this section we modify SDP-AE to give an $(\varepsilon, \delta)$-SDP algorithm for the MAB problem with improved additional regret. Recall that SDP-AE has a distribution-dependent regret of $O\left(\left(\sum_{a \in [k]:\Delta_a > 0}\frac{\log T}{\Delta_a}\right) + \frac{k \log \frac{1}{\delta}}{\varepsilon^2}\right)$, whereas the best known regret for $(\varepsilon, \delta)$ centralized-DP is due to Tossou and Dimitrakakis [28], and is $O\left(\left(\sum_{a \in [k]:\Delta_a > 0}\frac{\log T}{\Delta_a}\right) + \frac{k}{\varepsilon}\right)$. Since any SDP algorithm can be emulated by a centralized-DP mechanism, we can only wish to match the regret of Tossou and Dimitrakakis [28]. Hence, the natural question is: "Can we reduce the dependence in $\varepsilon$ of the additive (second) regret term from $\frac{1}{\varepsilon^2}$ to $\frac{1}{\varepsilon}$?"[15]

---

[15]Note that a Naive attempt to convert the algorithm of Tossou and Dimitrakakis [28] from the centralized model to the shuffle model fails. This is since in the centralized model they can decrease the relative noise as the number of iteration increases by keeping the raw data from the users and reusing it in later iterations. However, in our setting, the noise per shuffle batch has to remain constant for privacy, and we do not have access to the raw data of the users.

Before presenting our solution, we first indicate two intuitive approaches that fail. The first is to continue using batches of constant size $m = \Omega(1/\varepsilon^2)$. This idea fails since intuitively in the worst case we can expect a first batch regret of $m = \Omega(1/\varepsilon^2)$ which already surpasses $O(1/\varepsilon)$. The second is to try to directly adapt the algorithm of Tossou and Dimitrakakis [28], which can be interpreted as AE with batches of size $\lceil 1/\varepsilon \rceil$. Unfortunately, this adaptation fails since it requires that the added noise to the empirical mean of each arm decreases with time, whereas in the shuffle model, we must ensure privacy with respect to each batch equally and independently, so the total noise cannot decrease with time.

Now for our improved algorithm, observe that using a large batch size increases the regret a lot for arms which are very suboptimal. This is since rather than pulling these arms only a few times until we detect that they are suboptimal, we commit ourselves to pulling them throughout a large batch. On the other hand, fixing the batch size to be small increases the overall estimation error, since every application of the SDP summation algorithm $M_{sum}$ introduces an error of the same magnitude (specifically, the error is sub-Gaussian with variance $\sigma_{\varepsilon,\delta}^2 = O\left(\frac{\log\frac{1}{\delta}}{\varepsilon^2}\right)$) independently of the batch size. Hence, more executions of $M_{sum}$ translates to more noise due to privacy. We therefore extend SDP-AE to support variable size batches, which start small and gradually increase. Intuitively, the smaller batches initially, allow us to quickly eliminate very suboptimal arms with only a small number of pulls. We gradually increase the batch size to reduce the per-user error introduced by the private binary summation mechanism. Specifically, we double the batch size after each phase.

### 3.2.1 Algorithm outline

We consolidate the idea above into Algorithm 2 below, which we call *Variable Batch Shuffle Differentially Private Arm Elimination (VB-SDP-AE)*, which uses a different batch size $m^t = 2^t$ for each phase $t$:

---
**Algorithm 2:** VB-SDP-AE (Variable Batch Shuffle Differentially Private Arm Elimination)

---
1   **Input:** privacy parameters $\varepsilon$ and $\delta$ and horizon $T$.
2   **Initialize:** $\hat{S}_a^0 = 0$, $N_a^0 = 0$ and $\hat{\mu}_a^0 = 0$ for every $a \in [k]$;
3   Let $\sigma_{\varepsilon,\delta}^2 = O\left(\frac{\log\frac{1}{\delta}}{\varepsilon^2}\right)$ be the sub-Gaussian variance of the error distribution of $M_{sum}$;
4   Let $V = [k]$ denote the set of viable arms;
5   **for** *phase* $t \leftarrow 1, 2, \ldots$ **do**
6      Let $m^t \leftarrow 2^t$;
7      **for** *arm* $a \in V$ **do**
8          **for** *each new user* $i \leftarrow 1$ **to** $m^t$ **do**
9              User $i$ pulls the arm $a$ and observes reward $r_{a,i}^t$;
10              If total number of arm samples is $T$, exit;
11          **end**
12          **Communication:** Perform private binary summation $Z_a^t \leftarrow M_{sum}\left(\{r_{a,i}^t\}_{i=1}^{m^t}\right)$;
13          **Server update:** Update $\hat{S}_a^t \leftarrow \hat{S}_a^{t-1} + Z_a^t$, $N_a^t \leftarrow N_a^{t-1} + m^t$, and finally $\hat{\mu}_a^t \leftarrow \hat{S}_a^t / N_a^t$;
14      **end**
15      **Confidence bounds:** For each arm $a$, calculate $I_a^t \leftarrow \left(\frac{2\sqrt{t}\sigma_{\varepsilon,\delta}}{N_a^t} + \frac{1}{\sqrt{N_a^t}}\right) \cdot \sqrt{2\log T}$, and the upper and lower confidence bounds $UCB_a^t \leftarrow \hat{\mu}_a^t + I_a^t$ and $LCB_a^t \leftarrow \hat{\mu}_a^t - I_a^t$;
16      **Elimination:** remove all arms $a$ from $V$ such that $UCB_a^t < \max_{a' \in S} LCB_{a'}^t$;
17   **end**

---

### 3.2.2 Analysis

The privacy is trivial, since in each batch we use the $(\varepsilon, \delta)$-SDP mechanism $M_{sum}$. We now focus on regret.

To give a regret bound on VB-SDP-AE, we follow a similar proof to that of Theorem 3.1.

**Theorem 3.3.** *The algorithm VB-SDP-AE is* $(\varepsilon, \delta)$*-SDP, and has a distribution-dependent regret of* $O\left(\left(\sum_{a\in[k]:\Delta_a>0}\frac{\log T}{\Delta_a}\right) + k\sigma_{\varepsilon,\delta}\log T\right)$*, and a distribution-independent regret of* $O\left(\sqrt{kT\log T} + k\sigma_{\varepsilon,\delta}\log T\right)$.

*Proof sketch.* A slight modification of the proof of Theorem 3.1 lets us recover that with high probability, $\forall a \in [k], \forall t \in [T]$, we have that $\left|\hat{\mu}_a^t - \mu_a\right| \leq \left(\frac{2\sqrt{t}\sigma_{\varepsilon,\delta}}{N_a^t} + \frac{1}{\sqrt{N_a^t}}\right) \cdot \sqrt{2\log T}$. Assuming this high probability event, adapting the proof of Theorem 3.1 to our exponentially growing batch sizes gives that each arm $a$ is necessarily eliminated after we have already performed $O\left(\frac{\log T}{\Delta_a^2} + \frac{\sigma_{\varepsilon,\delta}\cdot\log T}{\Delta_a}\right)$ pulls of the arm $a$. Unlike the proof of Theorem 3.1, since the batch sizes grow as $2^t$, the additional pulls of the arm $a$ required to complete the last batch before elimination, adds only a factor of 2 to the total number of pulls to the arm. Hence, the regret of the arm $a$ is at most $R_a = O\left(\frac{\log T}{\Delta_a} + \sigma_{\varepsilon,\delta}\cdot\log T\right)$.

To get the distribution-dependent regret we sum the regret above over all arms. For the distribution-independent regret, we split this sum based on if the arm $a$ has $\Delta_a < \beta$ where $\beta = \Theta\left(\sqrt{\frac{k\log T}{T}}\right)$ or not, and bound each sum separately. Namely, bound the regret over arms with $\Delta_a < \beta$ by $\beta T$. $\quad\square$

We recall that the private binary summation mechanism $M_{sum}$'s error distribution is sub-Gaussian with variance $\sigma_{\varepsilon,\delta}^2 = O\left(\frac{\log\frac{1}{\delta}}{\varepsilon^2}\right)$, i.e., $\sigma_{\varepsilon,\delta} = O\left(\frac{\sqrt{\log\frac{1}{\delta}}}{\varepsilon}\right)$, and apply Theorem 3.3 to get the following corollary.

**Corollary 3.4.** *VB-SDP-AE is* $(\varepsilon, \delta)$*-SDP and has a distribution-dependent regret of* $O\left(\left(\sum_{a\in[k]:\Delta_a>0}\frac{\log T}{\Delta_a}\right) + \frac{k\sqrt{\log\frac{1}{\delta}}\log T}{\varepsilon}\right)$*, and a distribution-independent regret of* $O\left(\sqrt{kT\log T} + \frac{k\sqrt{\log\frac{1}{\delta}}\log T}{\varepsilon}\right)$.

## 4 Conclusion and future work

In this paper, we gave and analyzed differentially private algorithms for the MAB problem, closing the inevitable multiplicative $\Omega(1/\varepsilon^2)$ regret gap between the local model and the centralized model, by considering the (intermediate) shuffle model. Our algorithms are batched variants of AE, which use a private binary summation mechanism for the shuffle model as a building block. Compared to the non-private AE algorithm's regret, our first algorithm SDP-AE has an additive factor of $\frac{k\log\frac{1}{\delta}}{\varepsilon^2}$ using constant size batches, and our second algorithm VB-SDP-AE improves the additive factor to $\frac{k\sqrt{\log\frac{1}{\delta}}\log T}{\varepsilon}$ by using exponentially growing batches, which enable the early detection and elimination of very suboptimal arms.

A natural future work is to extend our results (i.e., the usage of a private binary summation mechanism for the shuffle model) to more general RL settings such as linear/contextual bandits or Markov decision processes. It would also be interesting to study whether our $\log T$ term in the additional additive regret factor can be shaved through a more sophisticated algorithm, or an alternative analysis. Finally, it is interesting to study whether there are more refined distribution-dependent regret bounds that depend on the KL-divergence as in Kaufmann et al. [20].

## Disclosure of Funding

This work is partially supported by Israel Science Foundation (grants 993/17,1595/19,1871/19), German-Israeli Foundation (grant 1367/2017), Len Blavatnik and the Blavatnik Family Foundation, the European Research Council (ERC) under the European Union's Horizon 2020 research and innovation program (grant agreement 882396), the Yandex Initiative for Machine Learning at Tel Aviv University.

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
