# Appendix

## A    Missing proofs from Section 3.1

*Proof of Theorem 3.1.* First, note that for any arm $a$, and phase $t$, we have $\hat{\mu}_a^t = \frac{\sum_{s=1}^t M_{sum}(\{r_{a,i}^s\}_{i=1}^m)}{N_a^t}$, where $N_a^t = m \cdot t$.

We define the *clean event* $C := \left\{ \forall a \in [k], \forall t \in [T] \ \left|\hat{\mu}_a^t - \mu_a\right| \le I_a^t \right\}$, where $I_a^t := \left( \frac{2\sqrt{t}\sigma_{\varepsilon,\delta}}{N_a^t} + \frac{1}{\sqrt{N_a^t}} \right) \cdot \sqrt{2 \log T}$ is a confidence bound interval. We now show that the clean event $C$ occurs with high probability, that is $P(C) \ge 1 - 4T^{-2}$, and after that assume the event $C$ to simplify our analysis.

Indeed, for each arm $a$, we imagine both a reward tape of length $1 \times T$, with each cell independently sampled from the distribution $D_a$ of arm $a$, and a private-binary-summation-error tape of length $1 \times T$, with each cell independently sampled from the distribution of (the additive) error of the private binary summation mechanism $M_{sum}$ for $m$ users.

We assume that in the $j$'th time a given arm $a$ is pulled by the algorithm, its reward is taken from the $j$'th cell in this arm's reward tape, and similarly the $j$'th time we compute a private binary sum over rewards of a batch of users who pulled $a$, the (additive) error is taken from the $j$'th cell in the arm's private-binary-summation-error tape.[16],[17]

Let $t \in [T]$ and $a \in [k]$, and let $\hat{v}_a^t$ be the approximated reward of arm $a$ that the algorithm would have held at the end of phase $t$ using the concrete values in the tapes defined above, that is $\hat{v}_a^t = \frac{\sum_{s=1}^t d_s + \sum_{i=1}^{N_a^t} e_i}{N_a^t}$, where $d_s$ is the $s$'th cell of the private-binary-summation-error tape of $a$, and the $\{e_i\}_{i=1}^{N_a^t}$ are the total $N_a^t = m \cdot t$ cells of the reward tape of $a$ that we have used until the end of the $t$'th phase.

Our aim is to bound the term $\left|\hat{v}_a^t - \mu_a\right| = \frac{\sum_{s=1}^t d_s + \sum_{i=1}^{N_a^t}(e_i - \mu_a)}{N_a^t}$. We first bound the first sum in the nominator, then bound the second sum in the nominator, and finally combine the bounds to get a bound for $\left|\hat{v}_a^t - \mu_a\right|$.

To bound the $d_s$'s sum, we apply Hoeffding's inequality (Lemma 2.1) for a sum of $n \leftarrow t$ random variables which are sub-Gaussian with variance $\sigma_{\varepsilon,\delta}^2$ and have zero mean (since $M_{sum}$ is unbiased) to get,

$$P\left( \left| \sum_{s=1}^t d_s \right| \le 2\sigma_{\varepsilon,\delta} \cdot \sqrt{2t \log T} \right) \ge 1 - 2\exp\left( -\left( 4\sigma_{\varepsilon,\delta}^2 \cdot 2t \log T \right) / \left( 2t\sigma_{\varepsilon,\delta}^2 \right) \right)$$

$$= 1 - 2T^{-4}. \tag{1}$$

To bound the $f_i = (e_i - \mu_a)$'s sum, we apply Hoeffding's inequality (Lemma 2.1) for the sum of $n \leftarrow N_a^t$ random variables $f_i$, each sub-Gaussian with variance $1/4$ (since it is bounded in the interval $[-\mu_a, 1 - \mu_a]$ of size 1), and with zero mean (since by its definition $\mathbb{E}[e_i - \mu_a] = \mathbb{E}[e_i] - \mu_a = 0$), to get that

$$P\left( \left| \sum_{i=1}^{N_a^t}(e_i - \mu_a) \right| \le \sqrt{2N_a^t \log T} \right) \ge 1 - 2\exp\left( -(2N_a^t \log T)/(2N_a^t/4) \right) = 1 - 2T^{-4}. \tag{2}$$

---

[16]Here we rely on the fact that the distribution of the error of $M_{sum}$ is independent of the input.

[17]Note that sizes of both tapes have been chosen conservatively to be of size $T$. We never pass the end of any these tapes, since there are at most $T$ users in total, and at most $T$ batches (actually roughly $T/m$ in this case), and we may not use them all.

Applying a union bound and the triangle inequality on Equation (1) and Equation (2) gives that

$$P\left(\left|\sum_{s=1}^{t} d_s + \sum_{i=1}^{N_a^t}(e_i - \mu_a)\right| \le \left(2\sqrt{t}\sigma_{\varepsilon,\delta} + \sqrt{N_a^t}\right) \cdot \sqrt{2\log T}\right) \ge 1 - 4T^{-4},$$

which by the definition of $\hat{v}_a^t$ and $I_a^t$ means that

$$P\left(\left|\hat{v}_a^t - \mu_a\right| \le I_a^t\right) \ge 1 - 4T^{-4}. \tag{3}$$

Since in the analysis above $t$ and $a$ are arbitrary, Equation (3) holds for every $t \in [T]$ and $a \in [k]$. Thus, we take a union bound over all arms $a \in [k]$ (assuming $k \le T$) and all $t \in [T]$, to conclude that

$$P\left(\forall a \in [k], \forall t \in [T] \left|\hat{v}_a^t - \mu_a\right| \le I_a^t\right) \ge 1 - 4T^{-2}. \tag{4}$$

Since the event in the probability above in Equation (4) is precisely the event that $C$ holds for a run of the algorithm using the randomness in the tapes as defined above (by the definitions of $\hat{v}_a^t$ and $\hat{\mu}_a^t$), we get that

$$P\left(C\right) \ge 1 - 4T^{-2}.$$

For the regret analysis, we assume the clean event $C$. Consider a suboptimal arm $a$ such that $\Delta_a = \mu^* - \mu_a > 0$, and consider the last phase $t_0$ following which we did not remove the arm $a$ yet (or the last phase if $a$ remains active to the end). Since we assumed the clean event, an optimal arm $a^*$ cannot be disqualified, and since $a$ is not yet disqualified, the confidence intervals of the arms $a$ and $a^*$ at the end of the $t_0$'s phase must overlap. Therefore,

$$\Delta_a = \mu^* - \mu_a \le 2(I_a^{t_0} + I_{a^*}^{t_0}) = 4I_a^{t_0} = \left(\frac{8\sqrt{t}\sigma_{\varepsilon,\delta}}{N_a^{t_0}} + \frac{4}{\sqrt{N_a^t}}\right) \cdot \sqrt{2\log T}, \tag{5}$$

where the third step follows since $a$ and $a^*$ were sampled using identical batch sizes throughout the algorithm, so at the end of the $t_0$'th phase, $N_a^{t_0} = N_{a^*}^{t_0}$ and therefore $I_a^{t_0} = I_{a^*}^{t_0}$, and the last step follows by the definition of $I_a^{t_0}$.

Observe that if $N_a^{t_0} > \frac{128\log T}{\Delta_a^2}$ then $\frac{4 \cdot \sqrt{2\log T}}{\sqrt{N_a^{t_0}}} < \frac{\Delta_a}{2}$, and if $N_a^{t_0} > \frac{(16\sigma_{\varepsilon,\delta} \cdot \sqrt{2\log T})^2}{m\Delta_a^2}$ then

$$\frac{8\sqrt{t_0}\sigma_{\varepsilon,\delta} \cdot \sqrt{2\log T}}{N_a^{t_0}} = \frac{8\sqrt{N_a^{t_0}/m}\sigma_{\varepsilon,\delta} \cdot \sqrt{2\log T}}{N_a^{t_0}} = \frac{8\sigma_{\varepsilon,\delta} \cdot \sqrt{2\log T}}{\sqrt{mN_a^{t_0}}} < 8\sigma_{\varepsilon,\delta} \cdot \sqrt{2\log T} \cdot \frac{\Delta_a}{16\sigma_{\varepsilon,\delta} \cdot \sqrt{2\log T}} = \frac{\Delta_a}{2},$$

so their sum is $< \Delta_a$ in contradiction to Equation (5). Hence, $N_a^{t_0} \le \max\left(\frac{128\log T}{\Delta_a^2}, \frac{512\sigma_{\varepsilon,\delta}^2 \cdot \log T}{m\Delta_a^2}\right)$.

Therefore the total regret on arm $a$ is

$$R_a \le \Delta_a \cdot (N_a^{t_0} + m) \le \max\left(\frac{128\log T}{\Delta_a}, \frac{512\sigma_{\varepsilon,\delta}^2 \cdot \log T}{m\Delta_a}\right) + m\Delta_a$$

$$\le \frac{128\log T}{\Delta_a} + \frac{512\sigma_{\varepsilon,\delta}^2 \cdot \log T}{m\Delta_a} + m\Delta_a, \tag{6}$$

where the first step follows since since the arm $a$ is eliminated following phase $t_0 + 1$ (or if $t_0$ is the last phase, then we finish and don't sample $a$ after it) of batch size $m$ and is subsequently never pulled, and the second step follows by previous bound on $N_a^{t_0}$ and since $N_a^{t_0+1} = N_a^{t_0} + m$. We sum up the regret over all arms, to obtain a bound for the total regret denoted by $R$:

$$R = \sum_{a \in [k]:\Delta_a > 0} R_a \le \sum_{a \in [k]:\Delta_a > 0} \left(\frac{128\log T}{\Delta_a} + \frac{512\sigma_{\varepsilon,\delta}^2 \cdot \log T}{m\Delta_a} + m\Delta_a\right).$$

To complete the analysis, we argue that the bad event in which $C$ does not hold contributes a negligible amount to the expected regret $R(T)$. Indeed,

$$R(T) = \mathbb{E}\left[R \mid C\right] \cdot P(C) + \mathbb{E}\left[R \mid \bar{C}\right] \cdot P(\bar{C})$$

$$\leq \sum_{a \in [k]:\Delta_a > 0} \left( \frac{128 \log T}{\Delta_a} + \frac{512\sigma_{\varepsilon,\delta}^2 \cdot \log T}{m\Delta_a} + m\Delta_a \right) + T \cdot 4T^{-2}$$

$$= O\left( \sum_{a \in [k]:\Delta_a > 0} \left( \frac{\log T}{\Delta_a} + \frac{\sigma_{\varepsilon,\delta}^2 \log T}{m\Delta_a} + m\Delta_a \right) \right), \tag{7}$$

where the first step follows by the law of total expectation, and the second step follows since the regret is at most $T$, and by the previous bound on $P(C)$.

Now for the distribution-independent bound, assume the clean even $C$, and let $\gamma > 0$ be a threshold whose exact value we will set later. We group the arms $a$ based on if $\Delta_a < \gamma$ or not, to get

$$R = \sum_{a \in [k] | \Delta_a < \gamma} R_a + \sum_{a \in [k] | \Delta_a \geq \gamma} R_a$$

$$\leq T \cdot \gamma + \sum_{a \in [k] | \Delta_a \geq \gamma} \left( \frac{128 \log T}{\Delta_a} + \frac{512\sigma_{\varepsilon,\delta}^2 \cdot \log T}{m\Delta_a} + m\Delta_a \right)$$

$$\leq T \cdot \gamma + \frac{(128 + 512\sigma_{\varepsilon,\delta}^2/m)k \log T}{\gamma} + mk,$$

where the first step follows from splitting the regret from before to two sums, the second step follows since in the first sum $\Delta_a < \gamma$ and since there are only $T$ samples in total throughout all arms, and in the second sum we apply Equation (6), and the final step follows since there are $k$ arms in total and since the elements in the sum satisfy $\Delta_a \in [\gamma, 1]$.

We balance the first two terms by defining $\gamma$ to be $\gamma = \sqrt{\frac{(128 + 512\sigma_{\varepsilon,\delta}^2/m)k \log T}{T}}$, so the total regret is: $R \leq 2\sqrt{(128 + 512\sigma_{\varepsilon,\delta}^2/m)k \cdot T \log T} + mk$. By a similar argument to Equation (7) conditioning on whether or not $C$ occurred, we get that the expected regret $R(T)$ satisfies

$$R(T) \leq 2\sqrt{(128 + 512\sigma_{\varepsilon,\delta}^2/m)k \cdot T \log T} + mk + T \cdot 4T^{-2} = O\left( \sqrt{\left(1 + \frac{\sigma_{\varepsilon,\delta}^2}{m}\right)kT \log T} + mk \right).$$

$\square$

## B   Missing proofs from Section 3.2

*Proof of Theorem 3.3.* We continue identically to the proof of Theorem 3.1, except the fact that we need the $t$'th index of each arm's private-binary-summation-error tape to contain an iid sample of the error of the private binary summation mechanism $M_{sum}$ for $m^t = 2^t$ users. By the definition of $M_{sum}$, which is sub-Gaussian with the same variance for any number of users (batch size), the application of Hoeffding inequality as in the proof of Theorem 3.1 still follows. We conclude that the *clean event* $C := \left\{ \forall a \in [k], \forall t \in [T] \; |\hat{\mu}_a^t - \mu_a| \leq I_a^t \right\}$, where $I_a^t := \left( \frac{2\sqrt{t}\sigma_{\varepsilon,\delta}}{N_a^t} + \frac{1}{\sqrt{N_a^t}} \right) \cdot \sqrt{2 \log T}$ occurs with high probability, that is $P(C) \geq 1 - 4T^{-2}$.

For the regret analysis, we assume the clean event $C$. Let $a$ be a suboptimal arm, and let $t_0$ be the last phase following which we did not remove the arm $a$ yet (or the last phase if $a$ remains active to the end). As in the proof of Theorem 3.1, we get that

$$\Delta_a \leq \left( \frac{8\sqrt{t_0}\sigma_{\varepsilon,\delta}}{N_a^{t_0}} + \frac{4}{\sqrt{N_a^{t_0}}} \right) \cdot \sqrt{2 \log T}. \tag{8}$$

We now diverge from the proof of Theorem 3.1.

Observe that if both $N_a^{t_0} > \frac{128 \log T}{\Delta_a^2}$ and $N_a^{t_0} > \frac{16\sqrt{t_0}\sigma_{\varepsilon,\delta} \cdot \sqrt{2\log T}}{\Delta_a}$, then we get a contradiction to Equation (8) since $\frac{4 \cdot \sqrt{2\log T}}{\sqrt{N_a^{t_0}}} < \frac{\Delta_a}{2}$ and $\frac{8\sqrt{t_0}\sigma_{\varepsilon,\delta} \cdot \sqrt{2\log T}}{N_a^{t_0}} < \frac{\Delta_a}{2}$ respectively. Hence, $N_a^{t_0} \leq \max\left(\frac{128 \log T}{\Delta_a^2}, \frac{16\sqrt{t_0}\sigma_{\varepsilon,\delta} \cdot \sqrt{2\log T}}{\Delta_a}\right)$. Therefore the total regret on arm $a$ is

$$
\begin{aligned}
R_a \leq 4\Delta_a N_a^{t_0} &\leq \max\left(\frac{512 \log T}{\Delta_a}, 64\sqrt{t_0}\sigma_{\varepsilon,\delta} \cdot \sqrt{2\log T}\right) \\
&\leq \frac{512 \log T}{\Delta_a} + 64\sqrt{2}\sigma_{\varepsilon,\delta} \cdot \log T,
\end{aligned} \tag{9}
$$

where the first step follows since arm $a$ is eliminated following phase $t_0 + 1$ (or if $t_0$ is the last phase, then we finish and don't sample $a$ after it) with batch size $2^{t_0+1} = 2 \cdot 2^{t_0} \leq 3 \cdot N_a^{t_0}$, and the third step follows since $m^{t_0} = 2^{t_0}$ and $t_0 \geq 1$, so $T \geq N_a^t = \sum_{s=1}^{t_0} m^s = 2^{t_0+1} - 2 \geq 2^{t_0}$ and therefore $t_0 \leq \log T$.

Similarly to the proof of Theorem 3.1 which uses Equation (6) to get the distribution-dependent bound, here we use the analogous Equation (9) to conclude that the distribution-dependent bound is

$$
R(T) = O\left(\sum_{a\in[k]:\Delta_a>0}\left(\frac{\log T}{\Delta_a} + \sigma_{\varepsilon,\delta}\log T\right)\right) = O\left(\left(\sum_{a\in[k]:\Delta_a>0}\frac{\log T}{\Delta_a}\right) + k\sigma_{\varepsilon,\delta}\log T\right).
$$

Now for the distribution-independent bound, similarly to the proof of Theorem 3.1, assuming the clean event $C$, for any $\gamma > 0$ it holds that the total regret

$$
R \leq T \cdot \gamma + \frac{512k \log T}{\gamma} + 64\sqrt{2}k\sigma_{\varepsilon,\delta} \cdot \log T,
$$

and specifically for $\gamma = \sqrt{\frac{512k \log T}{T}}$, the total regret $R \leq \sqrt{2048k \cdot T \log T} + 64\sqrt{2}k\sigma_{\varepsilon,\delta} \cdot \log T$. Similarly to the argument in Equation (7), conditioning on whether the clean event $C$ occurred or not, we conclude that the expected regret $R(T)$ satisfies

$$
R(T) \leq \sqrt{2048k \cdot T \log T} + 64\sqrt{2}k\sigma_{\varepsilon,\delta} \cdot \log T + T \cdot 4T^{-2} = O\left(\sqrt{kT \log T} + k\sigma_{\varepsilon,\delta}\log T\right).
$$

$\square$

## C Private binary summation mechanism for the shuffle model

In this section, for any $\varepsilon, \delta \in (0,1)$ and number of users, we give an $(\varepsilon, \delta)$-SDP private binary summation mechanism for the shuffle model, with an (additive) error distribution which is unbiased and sub-Gaussian with variance $\sigma_{\varepsilon,\delta}^2 = O\left(\frac{\log(1/\delta)}{\varepsilon^2}\right)$, and which does not depend on the input. Consider a group of $m$ users, each with a binary value $x_i \in \{0,1\}$, and the target is to calculate the sum $\sum_{i=1}^m x_i$. Our mechanism splits to two different internal mechanisms based on whether $m$ is "small" or "large". Intuitively, to ensure that we add noise which is roughly $\frac{1}{\varepsilon}$, when we have less

than roughly $\frac{1}{\varepsilon^2}$ users, each one adds several bits of noise, and when we have more than roughly $\frac{1}{\varepsilon^2}$ users, each one adds a single bit of noise with some bias. This mechanism is summarized below:

---

**Algorithm 3:** $(\varepsilon, \delta)$-SDP binary summation mechanism for $m$ users

---

**1** $\tau \leftarrow \frac{96 \log(2/\delta)}{\varepsilon^2}$;

**2**

**3** // Local Randomizer

**4** **Function** $E(x)$**:**

**5**    **if** $m \leq \tau$ **then**

**6**      **return** $(x, y_1, \ldots, y_p)$ where $\{y_j\}_{j=1}^p$ are iid $y_j \sim Bernoulli(1/2)$, and $p = \left\lceil \frac{\tau}{m} \right\rceil$;

**7**    **else**

**8**      **return** $(x, y)$ where $y \sim Bernoulli\left(\frac{\tau}{2m}\right)$;

**9**    **end**

**10**

**11** // Analyzer

**12** **Function** $A(z_1, \ldots, z_n)$**:**

**13**    **if** $m \leq \tau$ **then**

**14**      **return** $\sum_{j=1}^n z_1 - \left\lceil \frac{\tau}{m} \right\rceil \cdot m/2$;

**15**    **else**

**16**      **return** $\sum_{j=1}^n z_1 - \tau/2$;

**17**    **end**

---

**Theorem C.1.** *For any $m \in \mathbb{N}$, $\varepsilon < 1$ and $\delta > 0$, Algorithm 3 is $(\varepsilon, \delta)$-SDP, unbiased, and has an error distribution which is sub-Gaussian with variance $\sigma_{\varepsilon, \delta}^2 = O\left(\frac{\log(1/\delta)}{\varepsilon^2}\right)$ and independent of the input.*

*Proof.* We first prove that the mechanism is $(\varepsilon, \delta)$-SDP, and then prove the other claims.

Indeed, consider two neighboring inputs $X = (0, x_2, \ldots, x_m)$ and $X' = (1, x_2, \ldots, x_m)$. To ease on the analysis, we define the random variable $B$ to be the sum of all the random bits (i.e., $y$ or $y_1, \ldots, y_p$ depending on $m$) over all users in $X$. We define $B'$ identically with respect to $X'$.

We first claim that the sum $M^*(X) = B + \sum_{j=1}^m x_j$ of the shuffled reported bits is $(\varepsilon, \delta)$-DP.

Since $B$ is binomial in both regimes, by Chernoff bounds as in Theorem E.1 in Cheu et al. [9], for any $\delta > 0$ it holds that $P\left(|B - \mathbb{E}[B]| \geq \sqrt{3 \mathbb{E}[B] \log \frac{2}{\delta}}\right) < \delta$. Therefore, define $I_c = \left(\mathbb{E}[B] - \sqrt{3 \mathbb{E}[B] \log \frac{2}{\delta}}, \mathbb{E}[B] + \sqrt{3 \mathbb{E}[B] \log \frac{2}{\delta}}\right)$, and we get that $P(B \notin I_c) \leq \delta$ (and similarly for $B'$).

To show that $\frac{P(B=t)}{P(B'=t-1)} \leq e^\varepsilon$ for any $t \in I_c$, we split to the two regimes of $m$: the small $m \leq \tau$ regime, and the large $m > \tau$ regime.

**Small** $m \leq \tau$**:** In this case, $B \sim Binomial(\left\lceil \frac{\tau}{m} \right\rceil \cdot m, 1/2)$, so $\mathbb{E}[B] = \left\lceil \frac{\tau}{m} \right\rceil \cdot m/2$. For any $t \in I_c$, it holds that

$$
\begin{aligned}
\frac{P(B=t)}{P(B'=t-1)} &= \frac{2\mathbb{E}[B] - t + 1}{t} \leq \frac{\mathbb{E}[B] + \sqrt{3\mathbb{E}[B]\log\frac{2}{\delta}} + 1}{\mathbb{E}[B] - \sqrt{3\mathbb{E}[B]\log\frac{2}{\delta}}} \\
&\leq \frac{\tau/2 + \sqrt{\tau/2 \cdot 3\log\frac{2}{\delta}} + 1}{\tau/2 - \sqrt{\tau/2 \cdot 3\log\frac{2}{\delta}}} = \frac{1 + \sqrt{6\log\frac{2}{\delta}/\tau} + 2/\tau}{1 - \sqrt{6\log\frac{2}{\delta}/\tau}} \\
&= \frac{1 + \varepsilon/4 + 2/\tau}{1 - \varepsilon/4} \leq \frac{1 + \varepsilon/4 + \frac{\varepsilon}{4}}{1 - \varepsilon/4} = \frac{1 + \varepsilon/2}{1 - \varepsilon/4} \leq e^\varepsilon, \quad (10)
\end{aligned}
$$

where the first step follows since $B, B'$ are iid binomial with $\lceil \frac{\tau}{m} \rceil \cdot m = 2\,\mathbb{E}\,[B]$ trials of success probability $1/2$, the second step follows since $t \in I_c \Rightarrow t \geq \mathbb{E}\,[B] - \sqrt{3\,\mathbb{E}\,[B]\log\frac{2}{\delta}}$ (which is non-negative) and since $\frac{2\,\mathbb{E}[B]+1-t}{t}$ is a decreasing function of $t$ for $t \geq 0$, the third step follows since $\frac{x+\sqrt{ax}+1}{x-\sqrt{ax}}$ is a decreasing function of $x$ for $x > a$, where we take $a = 3\log\frac{2}{\delta}$ and $x = \mathbb{E}\,[B] \geq \tau/2 > a$, the fourth step follows by dividing the nominator and the denominator by $\tau/2$, the fifth step follows by the definition of $\tau$, the sixth step follows since $\varepsilon < 1$ so $\tau \geq 8/\varepsilon$, and the last step follows since $\frac{1+x/2}{1-x/4} \leq e^x$ for any $x \in [0,1]$. This concludes the case $m \leq \tau$.

**Large $m > \tau$:** In this case, $B \sim Binomial(m, \frac{\tau}{2m})$, so $\mathbb{E}\,[B] = \tau/2$. For any $t \in I_c$, it holds that

$$
\begin{aligned}
\frac{P(B=t)}{P(B'=t-1)} &= \frac{m-t+1}{t} \cdot \frac{\frac{\tau}{2m}}{1-\frac{\tau}{2m}} \leq \frac{m-\tau/2+\sqrt{\frac{3}{2}\tau\log\frac{2}{\delta}}+1}{\tau/2-\sqrt{\frac{3}{2}\tau\log\frac{2}{\delta}}} \cdot \frac{\frac{\tau}{2m}}{1-\frac{\tau}{2m}} \\[2mm]
&= \frac{m-\tau/2+\sqrt{\frac{3}{2}\tau\log\frac{2}{\delta}}+1}{\tau/2-\sqrt{\frac{3}{2}\tau\log\frac{2}{\delta}}} \cdot \frac{\tau/2}{m-\tau/2} \\[2mm]
&= \frac{m-\tau/2+\sqrt{\frac{3}{2}\tau\log\frac{2}{\delta}}+1}{m-\tau/2} \cdot \frac{\tau/2}{\tau/2-\sqrt{\frac{3}{2}\tau\log\frac{2}{\delta}}} \\[2mm]
&= \left(1 + \frac{\sqrt{\frac{3}{2}\tau\log\frac{2}{\delta}}+1}{m-\tau/2}\right) \cdot \frac{1}{1-\sqrt{6\log\frac{2}{\delta}/\tau}} \\[2mm]
&\leq \left(1 + \sqrt{6\log\frac{2}{\delta}/\tau}+2/\tau\right) \cdot \frac{1}{1-\sqrt{6\log\frac{2}{\delta}/\tau}} \\[2mm]
&= \frac{1+\varepsilon/4+2/\tau}{1-\varepsilon/4} \leq \frac{1+\varepsilon/4+\varepsilon/4}{1-\varepsilon/4} \leq e^\varepsilon,
\end{aligned}
\tag{11}
$$

where the first step follows since $B, B'$ are iid binomial with $m$ trials of success probability $\frac{\tau}{2m}$, the second step follows since $t \in I_c \Rightarrow t \geq \mathbb{E}\,[B] - \sqrt{3\,\mathbb{E}\,[B]\log\frac{2}{\delta}} = \tau/2 - \sqrt{\frac{3}{2}\tau\log\frac{2}{\delta}}$ (which is non-negative) and since $\frac{m+1-t}{t}$ is a decreasing function of $t$ for $t \geq 0$, the sixth step follows since $m-\tau/2 \geq \tau-\tau/2 = \tau/2$, the seventh step follows by the definition of $\tau$, the eighth step follows since $\varepsilon < 1$ so $\tau \geq 8/\varepsilon^2 \geq 8/\varepsilon$, and the last step follows since $\frac{1+x/2}{1-x/4} \leq e^x$ for any $x \in [0,1]$. This concludes the case $m > \tau$.

We therefore conclude that in both regimes of $m$,

$$
\forall t \in I_c, \quad \frac{P(B=t)}{P(B'=t-1)} \leq e^\varepsilon.
\tag{12}
$$

A dual argument shows that $\frac{P(B=t)}{P(B'=t-1)} \geq e^{-\varepsilon}$ using the fact that $t \in I_c$ so $t \leq \mathbb{E}\,[B] + \sqrt{3\,\mathbb{E}\,[B]\log\frac{2}{\delta}}$ and substituting in the value of $\mathbb{E}\,[B]$ as in the cases above.

We define $k = \sum_{j=2}^{m} x_j$ to be the true sum of the bits of $X$, and the true sum of the bits of $X'$ minus one. Therefore, for any $F \subseteq \mathbb{N}$ it holds that

$$
\begin{aligned}
P(M^*(X) \in F) &= P(M^*(X) \in F \wedge B \in I_c) + P(M^*(X) \in F \wedge B \notin I_c) \\
&\leq P(M^*(X) \in F \wedge B \in I_c) + P(B \notin I_c) \\
&\leq P(M^*(X) \in F \wedge B \in I_c) + \delta
\end{aligned}
$$

$$= \delta + \sum_{s \in F} P(M^*(X) = s \wedge B \in I_c)$$

$$= \delta + \sum_{s \in F} P(B = s - k \wedge B \in I_c)$$

$$= \delta + \sum_{s \in F} P(B = s - k \wedge s - k \in I_c)$$

$$\leq \delta + \sum_{s \in F} e^\varepsilon \cdot P(B' = s - k - 1 \wedge s - k \in I_c)$$

$$= \delta + e^\varepsilon \cdot \sum_{s \in F} P(M^*(X') = s \wedge s - k \in I_c)$$

$$\leq \delta + e^\varepsilon \cdot \sum_{s \in F} P(M^*(X') = s)$$

$$= \delta + e^\varepsilon \cdot P(M^*(X') \in F),$$

where the first step follows by the law of total probability, the third step follows since $P(B \notin I_c) \leq \delta$, the fourth step follows by the law of total probability, the fifth step follows by the definition of $M^*(X) = k + B$, the seventh step follows by substituting $t \leftarrow s - k \in I_c$ into Equation (12), and the eighth step follows by the definition of $M^*(X') = k + 1 + B'$. A similar dual argument uses the fact that $\forall t \in I_c$, $\frac{P(B=t)}{P(B'=t-1)} \geq e^{-\varepsilon}$ to show that $P(M^*(X') \in F) \leq \delta + e^\varepsilon P(M^*(X) \in F)$, and we conclude that $M^*$ is $(\varepsilon, \delta)$-DP.

To see that $M$ is $(\varepsilon, \delta)$-SDP, note that in our mechanism $M(X)$, given the number of users $m$, the total number of bits $U$ that the server receives is constant. Therefore, the shuffler's output is a random permutation of its input, which is of constant size. Thus, the shuffler's output's distribution is identical to the output distribution of the mechanism which first selects the number $s$ of ones in the shuffler's input where $s \sim M^*(X)$, and then post-processes the output $s$ by outputting a randomly shuffled binary vector with $s$ ones, and $U - s$ zeros. Since we have shown that $M^*$ is $(\varepsilon, \delta)$-DP, by post-processing arguments we conclude the shuffler's output is $(\varepsilon, \delta)$-DP, so $M$ is $(\varepsilon, \delta)$-SDP.

Now for the other claims, first recall that in both regimes of $m$, the output of the mechanism is of the form $z = B + \sum_{j=1}^m x_j - \mathbb{E}[B]$ where $B$ is the only source of randomness in the mechanism. Therefore, the mechanism is unbiased since $\mathbb{E}\left[z - \sum_{j=1}^m x_j\right] = \mathbb{E}[B - \mathbb{E}[B]] = 0$. In addition, the (additive) error of the mechanism which is precisely $B - \mathbb{E}[B]$, is obviously independent of the input $\{x_i\}_{i=1}^m$, and only depends on the natural parameters of the problem.

Finally, to see that the mechanism's additive error is sub-Gaussian with variance $O\left(\frac{\log(1/\delta)}{\varepsilon^2}\right)$, it suffices to show that $B$ is sub-Gaussian with variance $O\left(\frac{\log(1/\delta)}{\varepsilon^2}\right)$, since $B$ is the additive error shifted by a constant (this constant is $\mathbb{E}[B]$ and adding constants does not change the sub-Gaussian variance).

Indeed, recall that in both cases $B$ is binomial, where in the small $m \leq \tau$ case $\mathbb{E}[B] = \left\lceil \frac{\tau}{m} \right\rceil \cdot m/2 \leq (\tau + m)/2 \leq (\tau + \tau)/2 = \tau$, and in the large $m > \tau$ case $\mathbb{E}[B] = \tau/2 \leq \tau$ as well. Therefore, by Chernoff bounds as in Theorem E.1 in Cheu et al. [9], we get that for any $t > 0$, $P(B - \mathbb{E}[B] \leq t) < \exp\left(\frac{-t^2}{3\mathbb{E}[B]}\right) \leq \exp\left(\frac{-t^2}{3\tau}\right)$ and $P(B - \mathbb{E}[B] \geq -t) < \exp\left(\frac{-t^2}{3\mathbb{E}[B]}\right) = \exp\left(\frac{-t^2}{3\tau}\right)$, so by the equivalent definition of a sub-Gaussian variable, $B$ is sub-Gaussian with parameter $O(\tau) = O\left(\frac{\log(1/\delta)}{\varepsilon^2}\right)$. $\qquad\square$