# OpenReview forum: "Differentially Private Multi-Armed Bandits in the Shuffle Model"
_NeurIPS.cc/2021/Conference — NeurIPS 2021 Poster_

### Official Review · Reviewer_HQch · 2021-06-30

**Rating:** 6
**Confidence:** 5

**Summary:**

An interesting paper that utilizes the shuffle model to achieve low regret and differential privacy simultaneously.

**Limitations And Societal Impact:**

Yes, the authors adequately addressed the limitations and potential negative societal impact of their work.

**Main Review:**

This paper studies MAB under local differential privacy up to a trusted shuffler (the shuffle model). This privacy guarantee is a weaker notion than LDP and it requires the user to not repeat through the entire horizon, but it could relax the 1/eps^2 dependency on the additive regret to 1/eps. The algorithm is mostly the same as MAB under LDP where each reward is with a Bernoulli noise, but m rewards are shuffled and aggregated until they are observed by the algorithm, which makes the noise level much smaller than LDP. Setting m=2^t further improves the regret. The adjustment of the noise level borrows the lemma from the shuffle model. This manuscript is very well-written and the presentation is very clear. The intuitive description of the algorithm and the proof are also sufficient that they could be easily understood. The manuscript also properly credits the referred techniques and the related work, which helps the readers to understand a larger scale of the area. I would like to see this manuscript accepted, though I would not be too surprised if otherwise as the manuscript could still be improved in several ways.

1. I think Bernoulli bandits is quite a limit and it seems to be possible to give regret bounds to general sub-Gaussian arms. The concentration needed can again be borrowed from the shuffle model. This can improve the overall technical contribution as well (which is currently lacking). It might also be interesting to consider other relaxations such as allowing the users to repeat for some limited amount of times (which is critical for the shuffle model to make sense).

2. I believe that it would be helpful to at least implement the algorithm and compare it with other LDP/DP methods to see the effect of batch learning and reduced noise. I am also curious if m=2^t will be too large in practice and if so what could be the practical choice of the batch size.

Minor comments:

line 6 and other places: almost -> nearly (almost could mean w.p. 1)
line 138: move Section 2.3 to Appendix? I don't see them useful.
line 160: Is this a "challenge"? If so, elaborate on it.
line 220: delta dependence missing


**Time Spent Reviewing:**

4

---

> ### Author Response · Authors · 2021-08-09
> **Thanks for the review, please see comment below**
>
> 1. Bernoulli bandits is quite a limit and it seems to be possible to give regret bounds to general sub-Gaussian arms.
>
> Response to review:
>
> We chose to focus on the binary case in order to simplify the algorithms and to clarify our contributions. Our framework is applicable (with some modifications) also to sub-Gaussian arms by using more complex protocols for real-valued summation in the shuffle model, e.g., using the protocol of [1].
>
> 2. Consider other relaxations such as allowing the users to repeat for some limited amount of times?
>
> Response to review:
>
> We can achieve a guarantee with respect to returning users by applying the group privacy to our scheme.
>
> 3. Implement the algorithm and compare it with other LDP/DP methods.
>
> Response to review:
>
> The main focus of our paper is theoretical, and the main comparison criteria is regret.

---

> > ### Comment · Reviewer_HQch · 2021-08-31
> > **Thanks for the response**
> >
> > After reading the response and other reviews my review remains the same.

---

### Official Review · Reviewer_Kux9 · 2021-07-15

**Rating:** 7
**Confidence:** 3

**Summary:**

This paper studies the multi-armed bandit problem under the constraint of approximate differential privacy. Previously, multi-armed bandits have been studied in other models of privacy such as the centralized pure DP model, the centralized approximate DP model, and the local pure DP model. However, this is the first work to consider the multi-armed bandit problem under the shuffle model of privacy.

To be clear, the model that the authors consider is as follows. At each round $t$, a batch of $m$ random users are selected. An action (e.g. an ad) is then selected for each user. The reward for each user is then presented to a shuffle DP mechanism.

In terms of results, the authors prove a regret bound which is slightly weaker than the result one can obtain in the central DP model but significantly stronger than the result one can obtain the local DP model. In particular, it removes a multiplicative $1/\epsilon^2$ dependence that appears in the regret bound for the local DP model.

In terms of techniques, the authors combine some ideas from the bandit literature and the differential privacy literature. In particular, their algorithms are based on the non-private arm elimination algorithm and they show how one can use private binary summation from the shuffle DP literature to obtain a shuffle DP algorithm for MAB. The authors also give two differential algorithms. The first algorithm uses static batch sizes in each round. The second algorithm uses dynamic batch sizes and they show that this improves the regret.

**Limitations And Societal Impact:**

Yes

**Main Review:**

**Strengths**
- The paper studies the MAB problem under the shuffle approximate DP setting. This has not been studied before.
- The paper provides novel results.
- The results are obtained using a nice combination of techniques from the DP and MAB literature including private binary summation from the DP literature and the arm elimination algorithm from the MAB literature.
- The algorithms are fairly easy to understand.
- Overall the paper is written well and the ideas are clearly explained.

**Weaknesses**
- My main concern is that the paper does not have that much new technical ideas. Algorithm 1 mainly combined known ideas from the DP and MAB literature (in particular, private binary summation and arm elimination). Algorithm 2 tacks on the additional idea of using increasing batch sizes.


**Target Audience**
- The results will be of interest to researchers working on online learning and theoretical aspects of differential privacy.

**Time Spent Reviewing:**

3

---

> ### Author Response · Authors · 2021-08-09
> **Thanks for the review, please see comment below**
>
> 1. The paper does not have that much new technical ideas.
>
> Response to review:
>
> We are the first to consider and design online-learning algorithms in the shuffle model of differential privacy (part of our contribution is in fact to define this model).
>
> We consider a setting in which users arrive one-by-one. Whenever a user arrives, we need to choose an arm for the user to pull (on-line). However, in our model, the feedback from the users is not obtained immediately after the action was specified. Instead, the feedback only arrives after the current  batch of users is completed (and is computed privately, using the shuffle model of DP). The analysis then had to combine an analysis of MAB (Multi-Armed Bandits) with batches with a private summation shuffle protocol.
>
> A Naive attempt to convert the algorithm of Tossou and Dimitrakakis from the centralized model to the shuffle model fails since in the centralized model the relative noise decreases as the number of iteration increases. (In more detail, Tossou and Dimitrakakis keep the raw data from the users and reuse it in later iterations.) In our setting, the noise per shuffle batch has to remain constant for privacy, and we do not have access to the raw data of the users.
>
> To get our best results we had to throw in an additional idea of shuffling batches of increasing sizes. This allows to reduce the noise due to privacy as the batch size increases. This increase does not hurt the regret since we make it when we accumulate confidence in the quality of the arms.

---

> > ### Comment · Reviewer_Kux9 · 2021-08-30
> > **Thanks**
> >
> > I thank the authors for their response. I do think the problem is interesting, the results are nice, and the paper is generally well-written. I will be increasing my score for the paper.

---

### Official Review · Reviewer_dj8Y · 2021-07-15

**Rating:** 7
**Confidence:** 3

**Summary:**

The paper studies the classic multi-armed bandit problem in the shuffled model of differential privacy. Paper's first contribution is the definition of shuffled differential privacy (SDP) in the multi-armed bandit setting.  The paper's main contribution is an SDP algorithm with the same regret guarantees as the best-known multi-armed bandit algorithm in the central privacy model.

**Limitations And Societal Impact:**

Yes.

**Main Review:**

Originality:

a) The definition of shuffled differential privacy for sequential decision-making problems is interesting and novel and would inspire future privacy research for these problems.

b) The algorithms proposed by the authors combine known MAB algorithms and SDP mechanisms with minimal tweaking.

c) However, one key innovation is the variable batch size selection in the VB-SDP-AE algorithm, which allows the algorithm to enjoy the same regret as that enjoyed by algorithms in the central privacy model.

Presentation and Correctness of claims: The paper is well written and easy to follow. The analysis, too, is easy to follow and correct.

To summarize, I like this paper and recommend it be accepted.


**Time Spent Reviewing:**

4

---

> ### Author Response · Authors · 2021-08-09
> **Thanks for the review, please see comment below**
>
> Thank you for the kind review and nice words!

---

> > ### Comment · Reviewer_dj8Y · 2021-08-31
> > **Final Response**
> >
> > After going through other reviews and authors' responses to those, my assessment of the paper remains unchanged.

---

### Official Review · Reviewer_iPAH · 2021-07-21

**Rating:** 7
**Confidence:** 4

**Summary:**

This paper adapts the shuffle model of differential privacy to the multi-armed bandit setting, and aims to design a bandit algorithm which has privacy guarantees similar to that in the local model, while simultaneously having regret guarantees similar to the central DP model. The authors propose two arm-elimination style algorithms, which uses a private binary summation mechanism with optimal error guarantees as a subroutine, in order to obtain improved regret guarantees over the local model. In particular, they show how a simple, constant batch size policy already gets rid of the multiplicative dependence on $1/\epsilon^2$. Then, they show how to adapt this policy to exponentially increasing batch sizes, improving the additive dependence from $\approx k/\epsilon^2$ to $\approx k/\epsilon$.


**Limitations And Societal Impact:**

Yes, adequately addressed.


**Main Review:**

While the ideas and techniques considered in this paper are fairly standard, the results of this work seem nice. I found it surprising that almost the same regret guarantees are achievable in the shuffle model as in the central model, so I quite enjoyed reading this work.

Overall, I found the paper to be very well written. The exposition of the models considered in the paper, as well as the theoretical results and proof ideas, were very easy to follow. I especially enjoyed the section on the variable batch size which discussed two intuitive approaches which fail.

I am wondering what dependence on the parameter \delta is to be expected in the regret analysis. In Table 1, the Centralized $(\epsilon,\delta)$-DP result displays no (explicit) dependence on $\delta$, although I suspect this dependence is hidden perhaps in $\epsilon$? Do the authors believe that the $\sqrt{\log{\frac{1}{\delta}}}$ dependence on the additive term in their analysis is tight? Is this dependence the same as in the central model, or is this the main separation?

Additionally, could the authors elaborate on the $\log(T)$ term that appears in the additive $1/\epsilon$ term in their analysis? In particular, why does this term not appear in the central $(\epsilon,\delta)$-DP analysis? Could you provide any intuition why you would expect this term is not necessary?


**Time Spent Reviewing:**

7

---

> ### Author Response · Authors · 2021-08-09
> **Thanks for the review, please see comment below**
>
> 1. What dependence on the parameter \delta is to be expected in the regret analysis.? Is it the same as in the central model?
>
> Reponse to review:
>
> The regret dependence in delta for (eps,delta) centralized DP does indeed exist, but is given implicitly in their algorithm (see corollary 3.2 in Tossou&Dimitrakakis). The gap we do have is reflected by the logT term.
>
> 2. Elaborate on the log⁡(T) term. Why doesn’t this term appear in the central model?
>
> Response to review:
>
> The reason why in (eps,delta) centralized DP they can get rid of the additive logT term is that they decrease the additive noise with time, and evaluate a lagging estimate for the UCBs of the arms. In our case however, the estimate for each arm is updated through a shuffle model which must ensure privacy for each batch independently, and we add a noise of variance 1/epsilon^2 per batch. Since an action can have log T batches, the variance is  (log T)/epsilon^2 per action.

---

### Decision · Program_Chairs · 2021-09-28

**Decision:**

Accept (Poster)

**Comment:**

This paper proposes new algorithms for differentially private multi-armed bandits in the shuffle model. Their algorithms nearly match the results in the central model. The paper is well-written and the reviewers were positive.
I would suggest the authors incorporate the feedback from the reviewers in the final version. In particular, I found the discussion on the Naive attempt to convert the algorithm of Tossou and Dimitrakakis from the centralized model to the shuffle model to be useful, and adding some of this to the paper would make it better.
I am happy to recommend acceptance for this paper.

**Consistency Experiment:**

NeurIPS has a long history of experimentation. In 2014, NeurIPS ran an experiment in which 10% of submissions were reviewed by two independent committees to quantify the randomness in the review process. This year, we repeated a variant of this experiment to see how the quality of the review process has changed over time.  This paper was part of the experiment and was therefore assigned to two committees (consisting of reviewers, an Area Chair, and a Senior Area Chair) that reached independent decisions.  If both committees made the same recommendation, this recommendation was followed. If a single committee recommended acceptance, the paper was accepted (with the exception of a few cases in which the other committee identified what we considered a fatal flaw, e.g., an error in a key result).

Both committees reached the same decision: **Accept (Poster)**

The other committee assigned to the paper recommended **Accept (Poster)**.  You can find the other set of reviews, along with any follow up discussion with the authors here:
https://openreview.net/forum?id=WEwtAYNWyHh